Artificial light at night correlates with seabird groundings: mapping city lights near a seabird breeding hotspot

Heswall Ariel-Micaiah 1 ahes107@aucklanduni.ac.nz
Miller Lynn 2
McNaughton Ellery J. 1
Brunton-Martin Amy L. 1
Cain Kristal E. 1
Friesen Megan R. 3
Gaskett Anne C. 1
1 School of Biological Sciences, University of Auckland , Auckland , New Zealand
2 BirdCare Aotearoa , Auckland , New Zealand
3 Department of Biology, Saint Martin’s University , Lacey, WA , United States
Ward Eric
Electronic publication date: 2022 Oct 18
Publication date: 2022
Volume: 10
Electronic Location ID: e14237
Received 2022 Jul 14; Accepted 2022 Sep 23
Copyright: © 2022 Heswall et al.
Copyright year: 2022
Copyright holder: Heswall et al.
License: This is an open access article distributed under the terms of the Creative Commons Attribution License, which permits unrestricted use, distribution, reproduction and adaptation in any medium and for any purpose provided that it is properly attributed. For attribution, the original author(s), title, publication source (PeerJ) and either DOI or URL of the article must be cited.
License URL: https://creativecommons.org/licenses/by/4.0/

Keywords: ALAN, Seabirds, Light pollution, Light attraction, Groundings, Conservation

Funding: The authors received no funding for this work.

==============================
Artificial light at night (ALAN) is a growing conservation concern for seabirds, which can become disoriented and grounded by lights from buildings, bridges and boats. Many fledgling seabirds, especially Procellariiformes such as petrels and shearwaters, are susceptible to light pollution. The Hauraki Gulf, a seabird hotspot located near Tāmaki Makaurau/Auckland, Aotearoa—New Zealand’s largest urban city, with a considerable amount of light pollution and regularly documented events of seabird groundings. We aim to identify the characteristics of locations especially prone to seabird groundings. We used an online database of seabirds taken to a wildlife rescue facility by the public to map 3 years of seabird groundings and test for correlations between seabird groundings and the natural night sky brightness. We found that areas with lower amounts of natural night sky brightness and greater light pollution often had a higher number of seabirds grounded. Further, we identified important seasonal patterns and species differences in groundings. Such differences may be a by-product of species ecology, visual ecology and breeding locations, all of which may influence attraction to lights. In general, seabird groundings correlate with the brightness of the area and are species-specific. Groundings may not be indicative of human or seabird population abundance considering some areas have a lower human population with high light levels and had high amounts of seabird groundings. These findings can be applied worldwide to mitigate groundings by searching and targeting specific brightly lit anthropogenic structures. Those targeted structures and areas can then be the focus of light mitigation efforts to reduce seabird groundings. Finally, this study illustrates how a combination of community science, and a concern for seabirds grounded from light attraction, in addition to detailed animal welfare data and natural night sky brightness data can be a powerful, collaborative tool to aid global conservation efforts for highly-at-risk animals such as seabirds.

Introduction

As urbanisation expands into rural landscapes (Nechyba & Walsh, 2004) it can denaturalise the environment and encroach into the habitat of many animals (Sharma, Ram & Rajpurohit, 2011; Yeo & Neo, 2010). One aspect of urbanisation that can change animal behaviour is artificial lighting at night (ALAN) or light pollution. Light pollution can be from, but not exclusive to, street lighting, vehicles, security lights and buildings (Gaston & Bennie, 2014) and is associated with changes to animals and plants (Da Silva, Valcu & Kempenaers, 2015; Duarte et al., 2019). Light pollution affects the behaviours and movements of young offspring such as hatchling logger head turtles, Caretta caretta, in which the effect of light impairs their ability to move seaward (Lorne & Salmon, 2007). Some animals avoid lit areas (Bliss-Ketchum et al., 2016), but many studies document attraction to light pollution (Eisenbeis et al., 2009; van Langevelde et al., 2017). Songbirds on annual migration routes are frequently documented as attracted to the lights, which leads to collisions with buildings (Hudecki & Finegan, 2018) and cruise ships (Bocetti, 2011).

Another group of birds that are likely to be affected by light pollution are seabirds. Research documents the effect of light pollution which can coincide with high rates of seabird groundings: seabirds can be grounded by lights and collide with anthropogenic structures, potentially as a result of disorientation and attraction causing death (Rodríguez, Dann & Chiaradia, 2017; Troy et al., 2013). ALAN-related deck strike is also a common occurrence on fishing vessels and cruise ships (Bocetti, 2011; Glass & Ryan, 2013; Holmes, 2017; Merkel & Johansen, 2011; Ryan, Ryan & Glass, 2021).

The effects of light pollution on seabirds can include collisions i.e., with a fishing vessel or with an anthropogenic structure, groundings (when a seabird lands on the ground and is unable to take off), starvation, dehydration, and predation events which can lead to various injuries (Le Corre et al., 2002; Rodríguez et al., 2017a). Inexperienced seabird fledglings are the age group primarily grounded by light pollution, and groundings are especially common among burrow-nesting Procellariiformes (Rodríguez et al., 2017b). Procellariiformes can have acute vision and olfaction (Hayes, Martin & Brooke, 1991; Martin & Crawford, 2015; Nevitt, 2000). However, fledglings of burrow-nesting seabird species have underdeveloped vision, likely due to their lack of exposure to visual information while underground, making them more likely to be prone to disorientation from lights (Atchoi, Mitkus & Rodríguez, 2020; Mitkus, Nevitt & Kelber, 2018). Due to most Procellariiformes’ having acute vision and olfaction this generates sensitivity to sensory information which can result in them being uniquely vulnerable to sensory threats, including ALAN (Friesen, Beggs & Gaskett, 2017; Heswall et al., 2021).

ALAN-related groundings occur internationally with records in Hawaii (Rodríguez et al., 2015; Telfer et al., 1987), the Canary Islands (Rodriguez & Rodriguez, 2009), Maltese Islands (Laguna, Barbara & Metzger, 2014), Canada (Wilhelm et al., 2021) the United Kingdom (Syposz et al., 2018) and New Zealand (Whitehead et al., 2019). For example, over 1,500 Barau’s petrel (Pterodroma baraui), and over 650 Audubon’s shearwater, (Puffinus lherminieiri bailloni), were grounded on Réunion Island between 1996 and 1999, mainly attracted to streetlights (Le Corre et al., 2002). In Aotearoa’s/New Zealand’s Hauraki Gulf, near the major city of Auckland, almost 70 Buller’s shearwater (Puffinus bulleri) and flesh-footed shearwater (Ardenna carneipes) succumbed to deck strike from the lights of a single cruise ship near Te-Hauturu-O-Toi (Morton, 2018).

Collisions and groundings potentially due to light pollution are likely to be more frequent when abundant seabird breeding colonies (especially burrow-nesting Procellariformes) are adjacent to large urban cities. Northern New Zealand is a seabird hotspot with over a quarter of the world’s seabird species (Gaskin & Rayner, 2013). The Hauraki Gulf, near the north of Te Ika-a-Māui/the North Island, is a key breeding habitat of global importance with approximately 27 seabird species in a region covering 1.2 million hectares (Figs. S1A–S1C) (Auckland Council, 2012; Gaskin & Rayner, 2013; Whitehead et al., 2019). Within this area, there are many seabird colonies, both on land and on islands. Some species are relatively widespread with multiple colonies (e.g., Grey faced petrel, Pterodroma gouldi) whereas others are endemic and only breed on specific islands of the gulf. For example, the Buller’s shearwater, only breeds on the Poor Knights Islands/Tawhiti Rahi, while the NZ storm petrel (Fregetta maoriana) only breeds on Te-Hauturu-O-Toi and the black petrel, (Procellaria parkinsoni), only breeds on Aotea and Te-Hauturu-O-Toi (Gaskin & Rayner, 2013). Burrow-nesting Procellariiformes are the most-common type of seabird in this region (Gaskin & Rayner, 2013).

In addition to hosting rare and diverse seabird species of burrow-nesting seabirds, known for their propensity of groundings due to ALAN, the Hauraki Gulf is also the location of NZ largest and rapidly growing city: Auckland/Tāmaki Makaurau. The associated light pollution from the city likely threatens marine and native ecosystems (McNaughton et al., 2022). Many of the seabirds which breed in the Hauraki Gulf often fly over Auckland to reach their foraging grounds in the Tasman Sea (e.g., Cook’s petrels (Pterodroma cookii) (Gaskin & Rayner, 2013)). As they fly over, they may be at a high risk of disorientation and grounding due to ALAN. However, there has been little research on the characteristics of locations where seabirds around New Zealand and more specifically, in Auckland are most likely to become grounded. Considering the Hauraki Gulf is home to many endemic and vulnerable seabirds, it is important to determine whether there are grounding hotspots, and whether these correlate with the natural night sky brightness (an indicator of light pollution intensity).

Here, we characterise the location of seabird groundings, and test for correlations with natural night sky brightness. We do this by mapping grounding locations of seabird species that have been rescued and deposited at the local avian rehabilitation facility, BirdCare Aotearoa. We combine this with data on the natural night sky brightness to explore these aims: Aim 1: To map and identify locations of seabird groundings in the Auckland region.

Aim 2: Test for relationships between seabird groundings and natural night sky brightness.

Aim 3: Explore any other relationships (e.g., seasonal differences, species differences and urban vs rural) with effect of ALAN.

Materials and Methods

Study site

Auckland is New Zealand’s largest city with over 1.4 million residents (Auckland Council, 2018) sprawling over 560 km2 (Statistics New Zealand, 2013). The Auckland region is an isthmus surrounded by the Hunua Mountain Ranges in the south-east, and the Waitakere Mountain Ranges to the north-west, bordered by the Tasman Sea to the west and the Haruaki Gulf to the east (Fig. 1). The Auckland region is surrounded by two major bodies of water, the Tasman Sea to the west and the Pacific Ocean to the east.

Figure 1 Map of the major cities and suburbs of Auckland with the black/dashed line indicating urban/rural boundary.

Black star indicates CBD.

Study species

We included the seabird species brought to BirdCare Aotearoa which were found grounded across Auckland city: Cook’s petrel, grey-faced petrel, black petrel, Buller’s shearwater, fluttering shearwater (Puffinus gavia), common diving petrel (Pelecanoides urinatrix), white-faced storm petrel (Pelagodroma marina), sooty shearwater (Ardenna grisea) (Table 1). These are all burrow-nesting seabirds which have breeding sites around Auckland and the Hauraki Gulf (Figs. S1A–S1C) (Miskelly, 2013). These species were included in our analyses because they have also been recorded in the literature to have been susceptible to the effects of ALAN (Gaskin & Rayner, 2013; Holmes, 2017; Whitehead et al., 2019; Zissis, 2020). Seabirds species which were excluded included the Suliformes such as the Australasian gannet (Morus serrator), as well as the Charadriiformes such as the southern black-backed gulls (Larus dominicanus), and red-billed gulls (Chroicocephalus novaehollandiae scopulinus). As of yet, nothing has been recorded in the literature about those seabird orders and species being attracted to lights.

Table 1 Seabird species brought into BirdCare Aotearoa from 2018–20211.

Cases were included in our analyses when seabirds have been recorded in the literature to have susceptible to the effects of ALAN.

	Species name	Te Reo name	IUCN rank2	NZ conservation status2	Breeding population numbers3,4,5,6,8	Total patient number1	% released1	Time of year of grounded seabirds1	Fledgling months7,8	
Cook’s petrel	Pterodroma cookii	Tītī	Vulnerable	Relict	>300,0009	247	73%	December–May	February–March7	
Grey-faced petrel	Pterodroma gouldi	ōi	Least Concern	Not threatened	~300,0003	41	46%	Throughout	December–January8	
Black petrel	Procellaria parkinsoni	Tāiko	Vulnerable	Nationally vulnerable	~50004	38	58%	December–January; May–August	April–July8	
Fluttering shearwater	Puffinus gavia	Pakahā	Least concern	Relict	>100,0003,7	18	22%	January–April	January–February8	
White-faced storm petrel	Pelagodroma marina	Takahikare-moana	Least concern	Relict	>1,000,0003,6	7	29%	November–March	January–March8	
Buller’s shearwater	Puffinus bulleri	Rako	Vulnerable	Naturally uncommon	~78,0005	48	81%	May	April–May8	
Common diving petrel	Pelecanoides urinatrix	Kuaka	Least concern	Relict	>1,000,0003,7	6	33%	August; December–January	November–January8	
Sooty shearwater	Ardenna grisea	Hākoakoa	Near threatened	Declining	>20,000,0003,7	14	36%	October–February February–June	April–May8	
Notes:

1 The Wild Neighbors Database Project (2021).

2 BirdLife International (2021).

3 Miskelly (2013).

4 Bell (2013).

5 Friesen et al. (2021).

6 Southey (2013).

7 Gaskin & Rayner (2013).

8 Taylor & Rayner (2013).

9 Rayner et al. (2007).

Data collection

Due to Auckland being a large city covering over 560 km2 (Statistics New Zealand, 2013), we were not able to use the ideal systematic approach to find and collect dead or alive grounded seabirds. Therefore, for this study we used a citizen science approach using data from the wildlife rehabilitation medical database (WRMD; The Wild Neighbors Database Project, 2021). WRMD is a database designed for wildlife rehabilitators to record, analyse and manage the data of the animals which enter into their care/facility. Commencing in 2016, the database is now used in over 25 different countries and almost 1,000 rehabilitation centers (The Wild Neighbors Database Project, 2021). We extracted data on seabirds that had been found in the Auckland region (both urban and rural areas) and brought into BirdCare Aotearoa from January 2018 until September 2021.

GIS analysis

The physical address where the seabird was found was obtained using the WRMD (The Wild Neighbors Database Project, 2021). We then inputted the physical addresses into the search engine Nominatum to generate a GPS coordinate (longitude/latitude) which would be used in OpenStreetMap (OSM). OSM is an open source mapping system that relies on user data to create publicly available maps (OpenStreetMap, 2021).

This was conducted in R studio using ‘tmaptools’ and the function ‘geocode_OSM()’ (Tennekes, 2021). If a coordinate was not found for an address, then it was filtered for matched addresses until all the coordinates for the addresses were found. Once all the addresses and coordinates matched, they were plotted out onto a map using the packages ‘rJava’ and ‘OpenStreetMap’ (Fellows, 2019; Urbanek, 2009).

Map projections

OSM uses the Universal Transverse Mercator (UTM), as its projection, dividing the earth into an even grid creating less distortion at the poles and preserving distance. To obtain the minimum amount of distortion for our location in Auckland, New Zealand, we defined our zone as ‘Zone 1’. Since our coordinates are in longitude/latitude and not UTM, our points were converted into a Spatial Points Data Frame (SPDF) for plotting onto the map using the packages ‘sp’, ‘raster’ (Hijmans, 2021; Pebesma et al., 2012) and ‘ggplot2’ (Wickham, 2016). Once converted, dot plots and probability heatmaps were constructed to identify the grounding locations in Auckland for each individual seabird species as well as changes over time.

Map construction

To construct the probability heat maps we used the kernel density estimate (stat_density_2d; ggplot2). This enabled us to find the probability that a future point might occur in that area, given the current distribution of points per 5 km2.

We also mapped the predicted natural night sky brightness, using data obtained from continuous measurements between November 2015 to August 2017 of Auckland’s night sky brightness (McNaughton et al., 2022). The predicted natural night sky brightness is a measure of the night sky quality and is an index. The index used has values between 13–22 magSQM/arcsec2 (McNaughton et al., 2022). The higher the value, the greater the natural night sky brightness meaning that there is a smaller intensity of light pollution (McNaughton et al., 2022). Conversely a lower value corresponds with a lower natural night sky brightness meaning that there is a greater intensity of light pollution (McNaughton et al., 2022).

Locations

To test for a relationship between the predicted natural night sky brightness and seabird groundings, the map of Auckland was divided into 30 raster grid squares in 10 km by 10 km using the package ‘raster’. The average, minimum and maximum predicted natural night sky brightness for each grid square was extracted. The seabird fallout coordinate points were imported into the raster and then extracted for each grid square. The 10 km by 10 km grid squares were used to avoid zero inflated data – smaller grid length generate many grid squares with 0 seabirds.

An urban/rural delineation was generated using the urban boundary data from Auckland Council (2012). We used the addresses where the grounded seabirds were found in order to categorise the grounding into either a rural or urban area using the delineation. These categories were used in statistical analysis to look for relationships between seabird fallout and urban/rural areas.

Statistical Analyses

Seabird groundings, location and dates

To test whether the number of seabird groundings depended on natural sky brightness, we calculated the maximum brightness in a 10 km by 10 km grid. We fit one generalised linear models (poisson distribution) using the R package, ‘glmmTMB’, with number of groundings as the response variable and maximum brightness as the predictor (Brooks et al., 2017). We checked assumptions of the model using the R package, ‘DHARMa’ – as the data were zero-inflated, we included a formula for zero-inflation in the model. Goodness of fit was measured using McFadden’s pseudo-R2 (Hartig, 2022).

To investigate how location whether urban or rural (also proxy for human population) and date influenced the probability of each of the nine species being grounding, we performed a set of multinomial logistic regressions using the R package ‘nnet’ (Venables & Ripley, 2002). These models predict the probability that a seabird species is being grounded according to the corresponding location or season. Predictions are made in turn for each variable location, while all other variables are considered fixed. Goodness of fit for the model was assessed using McFadden’s pseudo-R2. Models were fit using three types of data: The date of the grounding (numeric; Folland & Karl, 2001)

The type of location of the grounding (binary: urban vs rural; Folland & Karl, 2001),

The austral (Southern Hemisphere) season (factorial: winter, spring, summer, autumn).

Model selection was performed using Akaike’s Information Criterion (AIC), where we chose the model with the lowest AIC, and effects estimated using type-III analysis-of-variance (‘car’ package) (Fox & Weisberg, 2018). In several instances, perfect separation occurred as seabirds were only ever recorded as grounded in Autumn 2018. In these cases, season was excluded and our final model predicted the probability of a species grounding from date and location. We calculated predicted probabilities for pairwise contrasts using the package ‘lsmeans’ (Lenth, 2016) in R.

Results

A total of 356 seabirds from eight different seabird species were brought into BirdCare Aotearoa from January 2018–December 2021 which were found grounded (Table 1). The central business district (CBD) was a grounding hotspot (Fig. 2), with fewer groundings reported in the rural areas with the exception of seabird groundings in brightly lit rural areas (Fig. 2).

Figure 2 The number and location of the seabird groundings across Auckland in 5 km2 for all species from 2018–2021.

Associations with natural night sky brightness

The lowest natural night sky brightness in 2016 occurs in the CBD area as well as around the industrial areas and north and west Auckland (Fig. 3A). There is a negative correlation between seabird groundings and maximum natural night sky brightness, such that there is more fallout in areas with more light pollution (lower values of maximum natural night sky brightness) (Fig. 4; Table 2) (note that, as previously mentioned, high values of natural night sky brightness actually means a lower intensity of light pollution). The majority of the seabird groundings occurs in areas with a lower value of max night sky brightness (Fig. 3B; Fig. 4; Table 2).

Figure 3 Heat map of the light pollution data in Auckland in 2016 (McNaughton et al., 2022).

(A) Overlayed with the locations of the seabird fallout of all species from 2018–2021. (B) Predictions indicate the predicted night sky brightness or night sky quality, the higher the number the greater the night sky quality (magSQM/arcsec−2).

Figure 4 The seabird fallout frequency in 10 km by 10 km grid squares against the mean predicted night sky brightness.

Note that high values of night sky brightness actually mean a darker sky with less Artificial Light at Night (ALAN).

Table 2 Results from three linear models models examining the relationship between seabird fallout and the mean, max and min predicted night sky brightness for every 10 km by 10 km.

Night sky brightness	Estimate	Standard error	T value	P value	
mean	−15.769	3.711	−4.250	0.0001 × 10−06	
max	−21.960	3.725	−5.895	0.0001 ×10−06	
min	−11.038	3.572	−3.090	0.0001 × 10−06	

Urban vs rural

Most seabird species were grounded similarly across both urban and rural areas (Table S1; P > 0.05). However there were some species such as the black petrel and grey-faced petrels which tended to be grounded in the rural areas more often compared to the urban area (Table S1; Fig. 5), whereas the Cook’s petrel had a greater chance of becoming grounded in the urban area (Table S1; Fig. 5).

Figure 5 The predicted probability (+/−SE) of each seabird species landing in a grounded or rural area.

Species specific groundings

Cook’s petrels were most often grounded close to the central business district (CBD), with some scattered across Auckland (Fig. 6A). The black petrel groundings are more broadly distributed with occasional groundings around rural coastal areas in both eastern areas to the highly lit western areas (Fig. 6B). The grey-faced petrels were also broadly scattered but there was a greater concentration of groundings near the rural western areas which were lit up (Fig. 6C). For the fluttering shearwaters, some grounding hotspots included areas north of the CBD and areas around highly lit western area (Fig. 6D). The white-faced storm petrels were rarely grounded but if so they were found in the CBD or in western suburbs (Fig. 6E). The Buller’s shearwaters groundings were restricted to south Auckland (Fig. 6F). The common diving petrels were grounded across central Auckland (Fig. 6G). The sooty shearwater groundings were also widely scattered, but with the majority by in the CBD, the western beaches and northern Auckland (Fig. 6H).

Figure 6 Locations of the groundings of the different seabird species across Auckland.

(A) Locations of the groundings of Cook’s petrels. (B) Locations of the groundings of the black petrels. (C) Locations of the groundings of the grey-faced petrels. (D) Locations of the groundings of the fluttering shearwaters. (E) Locations of the groundings of the white-faced storm petrels. (F) Locations of the groundings of the Buller’s shearwaters. Locations of the groundings of the common diving petrels (G) Location of the groundings of the sooty shearwaters (H) The count is the number of seabirds grounded in every 5 km.

Changes over time

The proportion of grounded seabirds varied according to season and species (Fig. 7). More Cook’s petrels are grounded during late summer/early autumn, with another smaller peak in late spring (Fig. 7). Black petrels and occasionally grey-faced petrels tend to become grounded later than the Cook’s petrels, in late autumn (Fig. 7).

Figure 7 The absolute number of seabird groundings from each species from January 2018–September 2021 including the seasons.

Lines under graph indicates seabird presence and the gold star indicates approximate fledgling dates. 1Imber, West & Cooper (2003), 2Bell & Sim (2005), 3Imber (1976), 4Bell, Bell & Bell (2005), 5Taylor (2013), 6Southey (2013), 7Miskelly (2013), 8Hedd et al. (2012).

From 2018–2021, overall the number of seabirds groundings have increased, especially in the CBD locations, but also the rural areas (Figs. 8A–8D).

Figure 8 The groundings of all seabird species in 2018 (A), 2019 (B), 2020 (C) and 2021 (D).

In the figure, n represents the number of seabirds grounded.

Discussion

Overall, we found a relationship between natural night sky brightness in Auckland and seabird groundings demonstrating that light pollution influences the number of seabird groundings. Groundings were common in the CBD and industrial areas, coinciding with a higher light pollution intensity and lower natural night sky brightness. We also documented that urban and rural regions near the coast (e.g., north Auckland) often have a high number of seabird groundings, again correlating with higher light pollution or low natural night sky brightness.

Internationally, light pollution tends to affect seabird fledglings, as they take their maiden flight (Rodríguez et al., 2017a, 2017b) and our data is consistent with this. The majority of the seabirds found grounded in Auckland and brought to BirdCare Aotearoa were fledglings, and many of the groundings occurred during the fledgling season (Table 1). This is similar to the previous finding that most of the seabirds which collided with the cruise ship near Te-Hauturu-O-Toi were fledglings (Morton, 2018) which nest and breed in the Hauraki Gulf area near to Auckland (Gaskin, 2012; Gaskin & Rayner, 2013).

Regarding the total number of grounded seabirds, we found that most species were grounded similarly in urban and rural areas but this depended on species. Rural areas tend to have less light pollution in general, with the occasional hotspot, since there is a lower density of human population in these areas (Falchi et al., 2019). This could lead to fewer groundings of seabirds since there is less light causing disorientation. We note, however, that in rural areas there are likely fewer people and therefore fewer reports of seabird groundings, underrepresenting the true numbers in the rural areas (Laguna, Barbara & Metzger, 2014). Hence, we recommend pairing our work with systematic approaches that sample rural and urban areas where possible.

Most species were grounded in urban and rural areas at a similar rate. Some species such as the Cook’s petrel was mainly grounded in urban areas. Whereas the grey-faced petrel and black petrel had a greater probability of grounding in the rural areas compared to urban areas. There are specific light pollution hotspots in those rural areas along the western beaches of Bethells, Muriwai and Piha which could increase the groundings in the rural areas for those seabird species. Therefore, it may not necessarily be that more seabirds are found grounded in areas with more people because there is more people to find them. Groundings maybe more indicative of light pollution rather than the human density or rural vs urban.

As well as some rural areas having some patches of light pollution, another possible explanation of seabird groundings occurring along rural areas for certain seabird species could be due to their breeding locations. For example, the grey-faced petrel breed around the western beaches of Bethells, Muriwai and Piha (Gaskin & Rayner, 2013). Future research to explore distance from breeding colony and groundings is required.

Considering those locations have some high levels of light pollution, those seabirds are likely to become grounded at those locations. This finding aligns with some previous research regarding Cory’s shearwater, Calonectris borealis, on Tenerife, the Canary Islands, where seabirds from colonies closer to light polluted areas were more likely to become grounded (Rodríguez, Rodríguez & Negro, 2015). It is also possible that light pollution from those rural locations are spilling into the areas of the breeding colonies (Garrett, Donald & Gaston, 2020) which could also influence seabird grounding.

Risk of ALAN: species differences

The difference in groundings between the species may not be indicative of seabird abundance. Some seabird species such as the Cook’s petrel and black petrel which have a smaller population sizes were actually grounded more often compared to other species with a larger population size such as the common diving petrel and white-faced storm petrel (Table 1). It is also important to consider that population data for different species might include estimates from different times and usingdifferent methodologies (Friesen et al., 2021; Gaskin & Rayner, 2013; Rayner et al., 2007).

During the breeding season, at night, some Cook’s petrels, including fledglings, fly across the Auckland isthmus from the islands of the Hauraki Gulf to the Tasman Sea to forage and then back across Auckland from the Tasman Sea to the Hauraki Gulf (Rayner et al., 2007; Rayner et al., 2008; Gaskin & Rayner, 2013). Since Cook’s petrels have to cross the Auckland isthmus, and the associated city lights, there may be a greater chance of them becoming disoriented and grounded when compared to the other seabird species. In comparison, the Buller’s shearwater and sooty shearwater migrate to the North Pacific (Guzman & Myres, 1983; Spear & Ainley, 1999; Warham, 1996). Therefore, since they migrate northward during the non-breeding season there is a smaller chance of them crossing over the Isthmus and being attracted towards the lights, however micro-migrations during breeding seasons where they cross over the Auckland Isthmus multiple times can occur where they could be vulnerable to groundings by lights (Gaskin & Rayner, 2013; Rayner et al., 2017; Whitehead et al., 2019).

Similar to Cook’s petrels, the majority of black petrels were grounded in the CBD, again suggesting light pollution is critical. However, there were also some groundings close to the western beaches of Auckland where there is less people but also had a low amount of natural night sky. Considering the black petrel migrates to Chile and South America during the non-breeding season (Cabezas et al., 2012) it seems unusual that a substantial number are grounded around Auckland, including along the Western beaches. However, black petrels often forage during the breeding season around the west and east coast of the North Island of New Zealand, off the continental shelves (Bell, Sim & Scofield, 2009). They may travel over the CBD en route to their foraging sites.

In general, the species that are rarely grounded in Auckland are those that are not commonly known to cross the Auckland isthmus for their foraging, breeding or migration (fluttering shearwaters, Buller’s shearwaters, white-faced storm petrels, common diving petrels and sooty shearwaters). Fluttering shearwaters tend to remain around their local breeding area after the breeding season (Whitehead et al., 2019). Fluttering shearwaters have been found to occasionally cross over to the Tasman Sea (Gaskin, 2013). Since they tend to remain close to their breeding colony, they could be less likely to interact with Auckland’s city lights. Similarly, Buller’s shearwaters are rarely grounded on land in Auckland although many were attracted to a single cruise ship in 2018 (Morton, 2018). Buller’s shearwater is so far only known to breed on the Poor Knights Islands and has even been documented since the late 1970s to migrate to northern Pacific areas during the non-breeding season (Friesen et al., 2021; Guzman & Myres, 1983; Nakamura & Hasegawa, 1979). Therefore, the Buller’s shearwaters are mainly avoiding the Auckland area and is less likely to become grounded by the lights.

White-faced storm petrels had few groundings, and only in the CBD and in west Auckland (Henderson). Once again, these areas had a greater amount of night sky pollution and therefore it suggests that light pollution/intensity is an important factor (McNaughton et al., 2022). Similar to the other seabird species which were rarely grounded in the Auckland region, the white-faced storm petrel migrates to the south pacific and South America during the non-breeding season (Imber, 1984). Therefore, this species is less likely to cross over the isthmus during the fledgling season and become susceptible to light attraction.

Common diving petrels also had few groundings. This species often migrates south east towards the Antarctic polar front (Rayner et al., 2017). Furthermore, common diving petrels forage locally within a 45 km radius of their burrow (Dunphy et al., 2020). As a result, common diving petrels are also not likely to cross over the Auckland isthmus regularly and therefore less likely to become grounded by the lights compared to other species. However, when they were grounded, they were found close to the CBD and the industrial areas which would have the greatest number of light sources (McNaughton et al., 2022), suggesting that light pollution is an important factor for this species.

The sooty shearwater also migrates to the north Pacific during the non-breeding season to forage rarely crossing over Auckland city (Shaffer et al., 2006; Spear & Ainley, 1999). Therefore, these seabirds are less likely to be attracted and grounded from light pollution. Overall, susceptibility to light pollution intensity could correlate strongly with the location of seabirds foraging, breeding and migration paths. More research down this path is required.

Changes over time

The number of grounded seabirds has changed over 4 years. Between 2018 and 2020, there was a decline in seabird groundings, but this decline reversed in 2021. The decline in 2019 and 2020 (Fig. 8) could be a result of the coronavirus (COVID-19) which resulted in a nationwide and then city-wide lockdowns (Henrickson, 2020). This could have caused more people to stay in their homes, so they were less likely to discover grounded seabirds, and there may have been less light pollution, though this was not measured.

The increase in grounded seabirds from 2020–2021 could result from many different, non-mutually exclusive factors. Auckland’s main lockdown ended on October 7th 2020 (New Zealand Government, 2021), leading to more people on the streets, and potential more probability of finding grounded seabirds. This uptick might also be due to an increase in Auckland’s light pollution over the years (McNaughton et al., 2022), which could result in more seabirds becoming grounded in Auckland. However, the increase could also result from a greater public awareness of seabird fallout and groundings in Auckland. Therefore, there is a greater chance of people reporting a grounding and taking it to the bird rehabilitation facility, BirdCare Aotearoa. Campaign efforts have been made with news media articles and radio interviews (Dexter, 2022). Other articles for the local magazines and articles for the Department of Conservation are currently being written.

Furthermore, regular patrols of the CBD were established in April 2022 by the first author of this article. The date of the first patrol was 14th April 2022 in which volunteers would patrol sections of the CBD 2–3 times a week. These patrols will continue in the CBD in the next years between March to May to establish a systematic survey of seabirds grounded in the CBD. Similar patrols were observed in other locations, including Hawaii and the Canary islands when more members of the public were alerted to seabird groundings (Miles et al., 2010; Rodríguez, Rodríguez & Negro, 2015; Travers et al., 2021). Thes patrols helps build awareness of the need for better conservation for seabirds grounded by light pollution.

Light pollution and sensory ecology

Along with differences in breeding, foraging and migration routes, another reason for species variation in groundings may be differences in their visual ecology. For example, black petrels were grounded at a much higher rate than common diving petrels. In addition to differences in migration and foraging locations affecting the seabird groundings, black petrels and common diving petrels also differ in their visual anatomy. Black petrels have a much larger eye socket volume relative to its body size compared to the common diving petrel (Heswall et al., 2021). Seabirds with relatively larger eyes may be more susceptible to light attraction. However, more research quantifying correlations between species-specific sensory anatomy and threats associated with visual signals is required. Previous work has shown such differences related to bycatch numbers with seabirds (Heswall et al., 2021).

Conclusions

Auckland’s increased economic growth, expansion of the human population, and increased light pollution (Bennie et al., 2014; Gallaway, Olsen & Mitchell, 2010; Jiang et al., 2017; McNaughton et al., 2022) follows a trend in cities across the globe (Czarnecka, Błażejczyk & Morita, 2021; Operti et al., 2018). Given the negative consequence for these vulnerable species, mitigation techniques should be employed to reduce the light attraction of seabirds. Some studies have shown that using shields over their lights can reduce seabird light attraction (Reed, Sincock & Hailman, 1985). Changing the type and colour of light could affect seabird attraction (Rodríguez, Dann & Chiaradia, 2017; Syposz et al., 2021). Some cities in the US have had great success getting buildings in the CBD to turn off lights at night during migration, effectively reducing fallout and groundings (Smith, 2009).

Our study suggests light pollution predicts groundings in specific locations, suggesting modifications to the lighting of specific structures and buildings could make a strong and positive improvement in avoiding bird collisions. This study also emphasises the benefit of diverse collaborations and community science. We show here the power of combining community awareness, databasing of wildlife species, and ecologists’ statistical modelling of seabird groundings with natural night sky brightness data can help provide information about locations where seabird groundings and why there is a higher seabird groundings in that location. Due to this collaboration with BirdCare Aotearoa involving some community science where people bring in grounded seabirds, we were able to correlate seabird groundings with light pollution which can aid in mitigation management, conservation and can be applied to cities across the world.

We identified the hotspots and characteristics of the areas associated with seabird groundings. The brightness of a location impacts seabird groundings, with more brightly lit areas having a greater chance of seabird groundings compared to a dimmer area which has a lower chance of seabird groundings. It is not necessarily urban vs rural or population sizes but more to do with the specific seabird species and the natural night sky brightness. This research can be used to identify areas which are more brightly lit resulting in more grounded seabirds. Particular buildings and other anthropogenic structures which are brightly lit could be identified and the lights could be modified to reduce the light pollution and, consequently, seabird groundings. This can be applied internationally to other countries and cities to reduce light pollution, keep natural dark areas, reduce seabird groundings, and help their conservation.

Supplemental Information

Supplemental Information 1 (A) Known locations of breeding colonies for the Cook’s petrel, black petrel and grey-faced petrel (B), fluttering shearwater, white-faced storm petrel, Buller’s shearwater (C) and the common diving petrel and the sooty shearwater.

Click here for additional data file.

Supplemental Information 2 Output of the pairwise contrasts between different seabird species and the location (rural vs urban). * And bold indicated significance. Pseudo R2 is 0.08.

Click here for additional data file.

Supplemental Information 3 R studio code.

Click here for additional data file.

We wish to acknowledge BirdCare Aotearoa and Iryll Findlay for providing us access to their comprehensive WRMD. I also wish to thank PhD student from the University of Oxford, Lynn Lewis-Bevan, who created the GIS code to map of the seabird fallout locations and helped with the GIS analysis. Joshua Dawes from the University of Auckland helped with the linear regression model for the GIS raster plots of the seabird fallout frequency and predicted night sky brightness. Scientific illustrator and artist Vivian Ward created the maps of Auckland and seabird breeding locations. Finally, we thank the community for finding and bringing injured birds to vets and BirdCare Aotearoa. We also thank the staff at BirdCare Aotearoa for their expertise and support.

Additional Information and Declarations

Competing Interests

Author Contributions

Data Availability

The authors declare that they have no competing interests.

Ariel-Micaiah Heswall conceived and designed the experiments, performed the experiments, analyzed the data, prepared figures and/or tables, authored or reviewed drafts of the article, and approved the final draft.

Lynn Miller conceived and designed the experiments, prepared figures and/or tables, provided access to the online database, and approved the final draft.

Ellery J. McNaughton conceived and designed the experiments, prepared figures and/or tables, provided access to the online database, and approved the final draft.

Amy L. Brunton-Martin analyzed the data, prepared figures and/or tables, authored or reviewed drafts of the article, and approved the final draft.

Kristal E. Cain analyzed the data, prepared figures and/or tables, authored or reviewed drafts of the article, and approved the final draft.

Megan R. Friesen analyzed the data, prepared figures and/or tables, authored or reviewed drafts of the article, and approved the final draft.

Anne C. Gaskett conceived and designed the experiments, analyzed the data, prepared figures and/or tables, authored or reviewed drafts of the article, and approved the final draft.

The following information was supplied regarding data availability:

The code is available in the Supplemental File.

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
