# Peer review of "Artificial light at night correlates with seabird groundings: mapping city lights near a seabird breeding hotspot"

_PeerJ, doi:10.7717/peerj.14237_

## Round 0.1 · original submission · Major Revisions

Three reviewers have looked at this paper; all appreciated the work you've put into this and think it's a good candidate for PeerJ. Reviewers 1-2 have included a number of comments that I think will improve the manuscript.

Reviewer 1 ·

Basic reporting

The paper presents interesting information about the grounding of eight burrow-nesting species in New Zealand. It is accompanied by professional figures and tables. There are many relevant references and some context provided. I believe, however, that readability could be improved in some places and more detailed hypotheses added. I suggest some changes in the attached document.

Experimental design

The manuscript presents original primary research.

Validity of the findings

The manuscript presents original primary research about timing and location of collected seabirds species around New Zealand. The main focus, however, is on the effect of night sky brightness on the location of collection of seabirds. It is assumed that birds are more likely to ground in areas where they are more likely to be collected by the public. I would argue, however, that a systematic approach must be undertaken to come to this conclusion. Therefore, I suggest changing the focus of the paper and include more detailed and clearer description of methods.

Additional comments

Major comments:

The manuscript presents the numbers of the collected burrow-nesting seabirds by the public in New Zealand. These species are well-known for their propensity to ground in lit-up areas and thus, are most likely to be collected by the citizens when affected by light pollution. The presented analysis reveals differences in timing and location (urban/rural) of eight grounded seabird species. The main focus of the manuscript is around the effect of night sky brightness (presented as a measure of light pollution) on the location of collected seabirds. It is assumed that birds are more likely to ground in areas where they are more likely to be collected by the public. I would argue, however, that a systematic approach must be undertaken to come to this conclusion. Studies by Rodriguez et al. (2014), Podolsky et al. (1998) and Ainley et al. (2001) indicate differences in number of dead grounded birds collected by the public and systematic searches. Dead birds are unlikely to be reported to the rescue centers, but may contribute to the number and location of affected seabirds by light pollution. Furthermore, more people are more likely to live in lit-up areas (as indicated by authors in L313-315), and humans are more likely to see grounded seabirds in bright places. Thus, I would be more careful with the wording regarding the dataset that has been obtained. I would also suggest changing the focus from the effect of artificial light intensity on the location of seabird grounding to status of the seabird grounding around New Zealand.

Introduction: The introduction has interesting information. It could be, however, improved by focusing on specific subject in each of paragraphs and by giving examples relevant to the study case. Furthermore, I would suggest presenting detailed hypotheses and relate your predictions to the previously presented studies.

L79 and 91 I would argue that we do not know if light pollution causes attraction or disorientation (or perhaps both) in seabirds. It is not clear if birds are coming to the light because there are mechanisms that drives them towards the light, or perhaps birds are accidentally coming towards light and once they are in the halo of the light, they become disoriented. It also may be that first birds are attracted and then disoriented in the light. I would suggest using words such as ‘effect of light on seabird’.

L82-85 Would injury not be a secondary effect? I suggest explaining here why seabird would not be able to take off again. I would also consider that seabirds do not have to collide with fishing vessel or anthropogenic structure. They can simply land on it or within boundaries of a town (e.g. garden) and become grounded. Thus, it is perhaps worth differentiate two types of effect of light on birds: grounding and colliding. Alternatively, you could use one word for effects of light on seabirds. Furthermore, grounding is preferentially used throughout the script. Accordingly to presented definition, none of the collected grounded birds collided with structure. It is, however, inconsistent with the description of collected birds since also injured birds were used for the analysis. Please, clarify definitions. Finally, there are two ways of description: ‘birds are grounded’ and ‘birds are being/becoming grounded’. I would suggest being consistent in use of the phrases.

Minor comments:
L23-24 repeated use of ‘especially’
L65-67 I would suggest re-writing this sentence as the mention about hatchling turtles is not clear.
L75 There are suggestions that songbirds are congregating in lit-up areas (e.g. Evans et al. 2007) and thus I think there could be some kind of mechanisms that drives songbirds towards light. Thus, I think that strikes are not only a result of high rates of migrating birds.
L77 I would argue that seabirds are commonly active during the day. Most Procellariiformes are active during the night only during the breeding season.
L90 I would drop “and poor”
L93 Could you explain why they are uniquely vulnerable to sensory threats?
L104 I would suggest dropping “disorientation from”
L107-108 Please, add a reference. I would actually argue that there are many more variables that might affect the numbers of grounded seabirds. For example, there are more groundings of Manx shearwaters recorded in Mallaig, next to a colony of about 70,000 pairs, comparing to small number of groundings recorded at the coast of Wales next to the largest colony on Skomer Island (c. 300,000 pairs). The coast of Wales in brightly lit-up, while Mallaig is a sole source of light pollution in a dark landscape.
L122 I would suggest changing to “In addition to hosting rare and diverse seabird species of burrow-nesting seabirds, known for their propensity to grounding due to ALAN…”
L123 change “home to” to “location of”
L128-129 Do you mean in general or specifically in this area?
L132 Why is it important to correlate it with natural night sky brightness? I understand it is an indicator of light pollution intensity. I would suggest explaining that to the reader at this point.
L136 Why do you focus only on the key seabird species?
L137-139 Could you please explain in more detail your hypothesis? How would they relate to the previous findings in the literature?
METHODS
L146 Can you specify that it is a mountain range?
L166 I would suggest explaining what kind of observation you undertook to classify species as susceptible to light pollution. Which are the species that has never been reported as grounded due to light pollution? It would be good to explain to the reader which species were previously reported grounding in lit-up areas (with relevant references). Perhaps this could be included under section 2.2.
L167 I would suggest adding Rodriguez et al. 2017a as a reference.
L167 What do you mean by ‘normally’?
L178-180 Please, consider re-writing this sentence and avoid repeating ‘using’.
L185-194 Add a reference to ‘rJava’, ‘sp’ and ‘ggplot2’
L203 Could you explain what is the spatial and temporal resolution of continuous measurement of the sky? Could you also explain how did you choose the values? Are these values an average over a year/season of fledging?
L204-209 The description of the sky quality index is not clear. For example, ‘very bright night sky’ in L207 sounds like a starry night. I would suggest changing to ‘higher intensity of anthropogenic light’.
L226 delete ‘our calculated’
L228 Could you please explain three linear models used or indicate that the explanation will follow in the next paragraphs?
L236 What do you mean by three different types of data?
L237 Did you use Julian date?
L236-238 Could you please explain better the hypothesis that made you include the indicated variables? It is especially not clear to me, why you included the season as a variable since we would expect bird groundings during their fledging season (regardless of the arbitrary season division).

RESULTS
L250 “tend to ground”
L251 Could you please indicate the CBD area on one of the maps?
L252 I would suggest combing figure 2 and 3, since you are describing position of grounded birds by using the artificial light intensity in rural area. A reader has to jump from one map to the other to figure out the position described in the text.
L255 I suggest changing to ‘The highest artificial light intensity”. The predicted natural sky brightness and intensity of light pollution are treated as the same thing, and they often bring confusion in the text. I suggest clearing it up and be consistent with the naming.
L259 I would suggest re-writing the sentence to “The highest number of collected birds by the public occurred in areas with higher artificial light intensity”
L264 I would suggest using ‘more likely to be collected”
L271 Please, decide either to use an abbreviation or the full name.
L276 Are the beaches lit-up? Could you please make this sentence a bit clearer?
L277 Is CBD a hotspot for all species or only fluttering shearwater?
L270-282 Please, avoid repetition of word ‘scattered’. You could use: ‘occasionally found’, ‘a few individuals were collected from’.
L287 autumn
L290 I would use overall instead of generally. Is it a significant increase?

DISCUSSION
L296 I would suggest starting discussion with more specific sentence about the findings. The relationship was found between night sky brightness in part of New Zealand.
L300 higher light pollution levels/intensity
L302 Could you please indicate the proportion between young and adult birds in different species in the results?
L313 I suggest avoiding repeating ‘also’.
L322-325 Could you justify this sentence by presenting analysis that include the density of population in mentioned areas?
L336 -337 This paragraph is a start for the subtitles 5.1, 5. 2 and 5.3. I would suggest moving it above the sub-title 5.1.
L356 Could you please explain what are micro-migrations?
L403 I would suggest avoiding general statements that are not supported by an appropriate analysis.
L407 I would suggest indicating at the beginning that the trend in number of collected birds has been changing.
L423-425 That is great news. Could you please state the exact date/year? Does your study include the records from the systematic survey?
L438-439 I suggest citing here Miles et al. 2010

Miles, W., Money, S., Luxmoore, R. and Furness, R. W. 2010. Effects of artificial lights and moonlight on petrels at St Kilda. – Bird Study 57: 244–251.

L470 I would suggest avoiding using phrases like ‘amount of ALAN’ throughout the text, as it is not clear whether you mean the number of lamps, light intensity, or some other measure of ALAN.

L443-475 These three paragraphs are conclusions of your study. Therefore, I suggest moving first two paragraphs from sub-section 5.3 and include them in conclusion. Furthermore, I would recommend making the text more concise as there are repetitions in these three sections.

L668 I would recommend adding coordinates.
Figure 3 I believe that the (a) and (b) indicators are mixed. (a) I understand that there was never a case of two birds landing in the same spot. Or would those points be on top of each other? If so, I would suggest making a map that would reflect the numbers of birds grounded in the same spot by making adequately bigger dots.
(b) I suggest avoiding repetition of ‘predicted’ and instead indicating that it is a kernel density estimate derived from night sky brightness measurement.

·

Basic reporting

This manuscript presents evidence of different characteristics of seabird fallout, specifically regarding light pollution levels, urbanization areas and time. It is a comprehensive study which encompasses 8 species along a three-year period. It highlights the contribution of citizen science and awareness programs to data collection and to inform on environmental management. As expected, and in accordance with the literature fallout is more severe in areas with increased light pollution. It is a well structured manuscript, that is very relevant at a regional scale and useful for local management. It also shows that urban and rural areas cannot be used solely for determining where groundings will occur, as even smaller areas with localized light in rural areas can lead to groundings especially if they are closer to breeding colonies or migratory paths. This is very important to the management of light pollution and does generate new lines of research regarding this global issue.

Overall the results are interesting and some differences between species can be clearly found. Curiously most species sampled did not present higher levels of fallout within urban areas, being similarly distributed between urban and rural areas. This is an interesting result and is discussed in the text, albeit it could be presented more in depth, as this point could be crucial for urban management.

The main figures could be improved slightly, and the presentation of the results needs to be clearer, especially regarding the species specific data. More care needs to be taken towards the writing to make the text more cohesive and remove syntax or grammar errors (which can be found throughout the text).

Background information may be too dense for a research paper and could be reduced.

Experimental design

The authors express well the research conducted, but should add in the introduction some reference to their expectations (i.e., less groundings in rural areas? more groundings according to higher levels of light pollution?).

I believe this manuscript is a good fit for PeerJ

Validity of the findings

The results need to be more clear regarding urban vs rural areas of groundings. The discussion is well structured but could be made clearer, especially the initial paragraphs, with a more in depth and structured discussion on the urban versus rural differences in groundings. Data is presented well with sufficient figures and tables to support it. I would add one table with intraspecies analysis of differences between rural and urban groundings (presented in Fig. 5).

Additional comments

Major changes
> introduction. It’s too long and too many general details. I feel it could be condensed into fewer paragraphs with the same information. For example the first three paragraphs are quite general and could easily be one. On the six paragraph, lines 114-118 could be shortened as well. The knowledge gap is well identified and well written (lines 128-132). Line 134, I would change ‘identify the characteristics of the locations’ with something like ‘characterize the locations of groundings’, and would reduce the specific detail from 136 (that could be in methods, and here present a more general description e.g. ‘key seabird species that have been rescued by organized public campaigns’. Line 138 Is it necessary to specify both natural night sky brightness and night sky quality?
> Methods. Line 151. I would remove the ‘most commonly taken’ as this was not the criteria. This is explained in the 2.3 paragraph.
Paragraph 2.5 has allot of information that could be in the supplemental material, I would suggest presenting a summary of this information in the main text.
> results. Paragraph 4.2 needs to be slightly rewritten. I would state that most species were found similarly across urban and rural areas (as it seems indicate in Fig. 5; it would be interesting to see a Table of the differences between rural and urban for the same species. Are they significant? Or are most species similarly likely to fall on either area?) This presentation of results also contrasts with the opening statement in discussion where ‘grounding were more common in the CBD and industrial areas’
> discussion. Line 298. Is it missing a lower before ‘predicted natural night sky brightness’? Also the authors mention suburban classification of the areas both in the methods (line 164) and in discussion (line 298) however this was not analysed, only urban vs rural correct? Please join into a single phrase lines 300 to 302. Rewrite lines 322-325, as it stands there are some syntax errors ‘not be a case of that’ or ‘which have not many people’. The following paragraph (line 327 to 330) is significant, I would suggest starting the discussion of the rural vs urban with this information, then progress into specifying that some rural areas like the Bethels etc beaches are well lit, so it makes sense for these species to ground there, and thus skewing the results towards rural areas. I would also add some reference to the spill over of light pollution from urban centres to natural or protected areas (Garrett et al 2019 Animal Conservation). These results regarding different area types is something which needs to be well discussed, as we are mostly expecting (and getting from the natural sky brightness data analysis) that birds ground in light polluted areas, which we associate with urbanization.


Minor changes
line 38. 'cities internationally' to something like 'be applied worldwide...'
line 40-41. Correct 'Those targeting' to 'Those targeted structures...'. Or rewrite the phrase, as it stands it is a little dense.
line 115/118/152/etc. Species names in italic and common names should be same throughout the manuscript.
Line 115 ‘whereas others are endemic’ (remove ‘an’)
Line 130-132. This is great!
Line 155. Table 1 is general for all species so I would remove it from the Ardenna grisea brackets and place it outside on its own.
Line 138-206-208-221. Avoid slashes. Is it necessary to have both terms of night sky quality and natural night sky brightness? If they are interchangeable choose one, ideally the one that is less contradictory less prone to be confused with a polluted sky. Write urban vs rural.
Line 226. ‘We used the calculated average…’
Line 231. ‘ nine species being grounded’
Line 236-7. Place the references at the end of the line (the date of the grounding (numeric; REF).
Line 237. ‘the type of grounding location (binary: urban or rural)’
Line 298. ‘higher light pollution and lower? Predicted natural night sky brightness’ missing a lower?
Line 318. Remove the For example. Both species mentioned are the two species with the most clear difference between urban and rural groundings. For example should be mostly used to give information on situations outside the manuscript research.
Line 339. ‘The difference in groundings’
Line 403-404. This is great!
Line 407. ‘but the decline’. Also add the cross reference to Fig. 8 at the end of the sentence.
Line 454. Light pollution or ALAN, choose one term.

Note. Please review the text as there are many typos and small syntax mistakes that need to be corrected. I have pointed out a few (above) but take care for the whole document and identify where other errors may be. Be consistent with the terms used across the text.

Figures and tables

Fig. 4. Please explain the colour bar (I would link it with the Note on the values of night sky). And some text at the ends of the colour bar could be added for a more immediate interpretation, something like light polluted (around the 18 PNSB) > dark sky (21 PNSB)
Fig5. X axis is not location. Species? Also I would suggest put the name of the species in the axis itself, as for example grey faced petrel data for urban areas can be easily missed. Black petrel does not have any urban fallout data? There is nothing in the graph.
Fig7. season names could have a line delineating the span, so its clearer where it starts/ends. Right now does summer start in December and end in January (sorry northern hemisphere); If its possible space-wise, the legend could be a one column list.

references: check if Rodriguez 2017 are all correct. 3 refs only two have a and b.

·

Basic reporting

Basic reporting is generally acceptable but I would suggest eliminating the short discussion of non-seabird issues with lighting, where references are weak, and instead expanding a bit on the lighting index which is not explained but which is the crux of the paper. The paper could use some copy editing but is basically clear. Specific comments are in the manuscript. It looks as though some of the table captions have missing words

Experimental design

This is an interesting investigation that takes advantage of data on seabird stranding and light-index distribution to generate new and useful information. While it may be technically acceptable to refer all questions about the lighting index to a previous paper, I think this paper would be improved, especially the suggestion that its findings could be generalized, by a short discussion of the index itself.

I think it would be useful, when examining timing of strandings for different species, to note when those species are actually present in the area - ie distinguish between birds present and no strandings, and birds not present and non strandings.

Validity of the findings

The authors do a good job of working with data collected for purposes other than answering the questions posed in the paper. While it is intuitive that more birds are found where lights are brightest, to me the most interesting finding related to species that are 'required' to make frequent flights across the isthmus.

---

## Round 0.2 · accepted · Accept

Thanks for addressing the comments from the reviewers, especially Reviewer # 1; I think the manuscript is improved.

The last thing I wanted to double check -- per a reviewer's comment -- was that people who contributed to the paper and met authorship guidelines were not excluded from being authors (specifically, Lynn Lewis-Bevan, Joshua Dawes)? PeerJ uses the ICMJE authorship guidelines, linked here: https://www.icmje.org/recommendations/browse/roles-and-responsibilities/defining-the-role-of-authors-and-contributors.html#two